# Mental health in gay, lesbian and bisexual medical students

**Felipe Scalisa, Renata Kobayasi, Milton A. Martins** (ORCID) **\*, Patricia Tempski**

Center for Development of Medical Education and Department of Medicine, School of Medicine of the University of Sao Paulo, Sao Paulo, Brazil

\* mmartins@usp.br

## Abstract

Our study aimed to assess depression symptoms among lesbian, gay and bisexual medical students and the associations between these data and sociodemographic characteristics, quality of life, resilience and internalized stigma scores. A multicenter cross-sectional study was conducted using an online questionnaire. We used Beck Depression Inventory, Trait-State Anxiety Inventory, Internalized Homophobia Inventory and Brief Resilience Inventory to assess depression and anxiety symptoms, internalized stigma and resilience, respectively. We used the snowballing technique. The initial sample comprised five individuals known to the research group who, in turn, were asked to recommend an additional five participants from any medical school in the state of São Paulo, Brazil. Among the medical students recruited through a snowball strategy, 404 (55.6%) responded to the survey. We used multinomial logistic regression models, both crude and adjusted for gender and sexual orientation, to examine the associations between depression symptoms and other data. Among the students surveyed, 62.3% identified themselves as men, and 35.8% of men identified themselves as bisexual; 63.4% of the students presented symptoms of depression, including 70.2% of the women, 58.6% of the men (P = .015), 70.6% of the bisexual students and 58.8% of the lesbian or gay students (P = .032). Medical students with moderate to severe depression symptoms had lower mean quality of life scores than those with mild symptoms and those without symptoms (P < .001). Similar patterns during medical school were observed for quality of life and resilience scores (P < .001 for all comparisons). The internalized stigma scores followed a similar trend, with higher scores associated with more severe depression symptoms than with mild symptoms or no symptoms (P = .004). The percentage of gay, lesbian and bisexual medical students with depression and anxiety symptoms is high, especially among bisexual students and women. Increased internalized stigma, lower resilience, and poorer quality of life are associated with higher depression scores.

## Introduction

### Mental health among medical students

The well-being of medical students is a prominent concern in medical education. A high prevalence of depression and anxiety [1, 2], burnout symptoms [3], and suicidal thoughts [4] have

**Data Availability Statement:** The dataset supporting the conclusions of this article is included within the article and supporting file.

**Funding:** Conselho Nacional de Desenvolvimento Científico e Tecnológico (CNPQ), Brazil, (Grant

number 301526/2019-2 to MAM) and Fundação de Amparo à Pesquisa do Estado de São Paulo (FAPESP), São Paulo, Brazil (Grant number 2019/13850-9 to MAM and PT). The funders had no role in study design, data collection and analysis, decision to publish, or preparation of the manuscript.

**Competing interests:** The authors have declared that no competing interests exist.

been reported among medical students. Additionally, mental health challenges in medical students are associated with adverse outcomes, including lower academic performance [5], low levels of professionalism [6], reduced empathy [7, 8], less attentive care and health services for patients [9], and decreased success in post university medical careers [10].

The prevalence of mental health issues among medical students surpasses that reported in the general population of the same age and among university students overall [3, 4]. A lower quality of life and reduced resilience have emerged as significant factors associated with depression symptoms in medical students [11, 12]. Despite controlling for sex, age, social conditions, and the university environment, medical students have a lower quality of life than the general population [12]. High levels of harassment within the medical school environment may also contribute to these findings [13]. Simultaneously, resilience is conceptualized as a system of coping that depends on individual and collective attributes, including psychosocial support [14]. This characteristic poses a structural challenge within medical schools, where the atmosphere is often characterized by intense competitiveness and the stigmatization of individuals who seek assistance [15].

## Mental health among LGBT medical students

LGBTQIA+ (lesbian, gay, bisexual, transgender, queer, intersex, and asexual) individuals constitute a minority group in society, and LGBTQIA+ people in different contexts exhibit worse mental health outcomes than their heterosexual and cisgender peers [16]. Meyer [17] termed this phenomenon "minority stress," which is defined as the specific and chronic stress experienced by stigmatized groups, primarily stemming from the structural stigma imposed upon them by society. Consequently, individuals within the LGBTQIA+ community have increased susceptibility to depression and anxiety [18], as well as increased rates of alcohol and drug abuse [19], smoking [20], self-harm [21], and suicide [22], compared with the general population.

The internalization of stigma is widely recognized as a significant contributor to minority stress within the LGBTQIA+ population [23]. Internalized homophobia, biphobia, and transphobia can profoundly affect an individual's overall behavior [24] and are intricately linked to the development of certain personality traits, notably perfectionism and competitiveness, in academic and professional spheres [25, 26]. Internalized stigma is linked to reduced resilience and the perception of a low quality of life [27].

Recent research has indicated that LGBTQIA+ individuals in university settings have a 1.5 to 4 times greater prevalence of depression than their non-LGBTQIA+ counterparts [28–30]. Considering the intersection of Meyer's minority stress model with studies on medical education, LGBTQIA+ medical students may have an even greater incidence of mental health problems.

## Objective

Research on the mental health, well-being, quality of life, and academic environment of medical students has grown significantly in recent years [31]. However, studies focusing specifically on the mental health of LGBTQIA+ individuals taking medical courses remain limited. The LGBTQIA+ population is particularly susceptible to mental health challenges [17]. Recognizing and understanding the prevalence of these challenges is crucial for mitigating mental health issues and establishing effective support programs.

Recently, the Brazilian LGBT+ Health Survey showed that, in Brazil, episodes of discrimination and depression are high among the LGBT+ population and highlighted the challenge for health professionals to provide health care to this population [32, 33]. Concerning health

professionals in training, not much data exist on the mental health of gay, lesbian and bisexual students.

The objective of this study was to assess the prevalence of depression symptoms among gay, lesbian and bisexual medical students and to examine the associations between these symptoms and sociodemographic characteristics, quality of life, resilience, and internalized stigma scores.

## Materials and methods

### Study design and sample

Our study had a cross-sectional design and employed self-report questionnaires to assess the mental health, minority stress, quality of life, and resilience of self-identified gay, lesbian, bisexual, and transgender medical students in the State of São Paulo, Brazil. Our study was performed according to the STROBE guidelines.

### Ethics statement

The study received approval from the Research Ethics Committee of the University of São Paulo (protocol number 3.156.924). Prior to participation, all individuals signed informed consent forms, and their anonymity was ensured throughout the study.

### Data collection

The data were collected from March 1 to July 31, 2019 using an online survey platform. The snowballing method, considered the gold standard for studying stigmatized subgroups, was employed for participant recruitment [34]. This approach is widely recognized and frequently used for recruiting participants from large LGBTQIA+ populations [35, 36]. Participants were contacted and recruited primarily through social media platforms. The initial sample comprised five LGBTQIA+ individuals known to the research group who, in turn, were asked to recommend an additional five participants from any medical school in the state of São Paulo. We finished data collection when the invited medical students had already been invited by another colleague.

### Instruments

We assessed gender identity (men, women, other), sexual orientation (gay, lesbian, bisexual, and other), age, religious status, living arrangements (with family members, alone, or in shared housing with friends), participation in student associations, whether the participant resided and studied in the capital or a smaller city, university administration model, year of medical school, use of psychiatric medications, alcohol habits, tuition scholarship status, and self-perception of academic performance. When the volunteers answered "other" to gender identity or sexual orientation questions, they were asked to specify their answers.

We used the Beck Depression Inventory [37], comprising 21 questions scored on a scale ranging from 0 to 3, to evaluate each participant's level of depression (minimal, mild, moderate, and severe depression). The instrument used was the translated and validated version of Brazilian Portuguese [38], and it demonstrated internal consistency, with a Cronbach's alpha of 0.876 in our study.

For the assessment of anxiety symptoms, we used the Trait-State Anxiety Inventory (STAI-T) [39], comprising 40 questions scored on a scale ranging from 1 to 4 to denote mild, moderate, or severe anxiety levels. The instrument used was the translated and validated

version of Brazilian Portuguese [38, 40]. The IDATE-Trait subscale had a Cronbach's alpha of 0.928, whereas the IDATE-State subscale had a Cronbach's alpha of 0.931 in our study.

To assess internalized stigma, we utilized the Internalized Homophobia Inventory (IHNI), originally developed by Meyer (1995) [41] and adapted by Mayfield (2001) [42]. The IHNI comprises 23 questions scored on a scale ranging from 1 to 5, and the instrument had a Cronbach's alpha of 0.885 in our study.

To evaluate the quality of life, we employed two questions: one asking participants to assign a grade for their overall quality of life from 0 to 10 and another regarding their quality of life in medical school [7, 11].

Additionally, we administered the Brief Resilience Scale by Wagnild and Young (RS-14) [43], which consists of 14 questions scored on a scale ranging from 1 to 7. The instrument had a Cronbach's alpha of 0.90 in our study. Higher values of the questionnaire correspond to greater resilience. This questionnaire was translated into and validated in Brazilian Portuguese [44].

## Statistical analysis

Descriptive statistics were used to determine the distribution of the students' characteristics (means, standard deviations, frequencies). T tests or ANOVAs with Bonferroni post hoc correction were used for continuous variables, and chi-square tests were used for categorical variables. We categorized the Beck Depression Inventory score according to the participants' degree of depression symptoms (0–13 minimal depression, no depression, 14–19 mild depression, 20–28 moderate depression and 29–63 severe depression) and the Trait-State Anxiety Inventory score into mild, moderate or severe symptoms if the score was less than 33, between 33 and 49 or greater than 49.

We used multinomial logistic regression models (crude and adjusted for gender and sexual orientation) to study the associations between students' sociodemographic characteristics and the other parameters according to the clinical category of the Beck Depression Inventory.

We established the level of statistical significance as 0.05. The analyses were performed using IBM SPSS Statistics Version 22.

## Results

Among the 727 students contacted using the snowballing method, 404 (55.6%) responded to the questionnaire. Only three participants identified as transgender, and one identified as pansexual, making it impractical to generate statistically significant data for these specific populations. The reasons cited for declining to participate included a lack of time and emotional vulnerability.

Table 1 presents general information about the subjects included in our study. The average age was 22.7 years (SD 2.7). Of the participants, 249 (62.3%) were men, with 37 (14.9%) identifying as bisexual. Among the 151 women (37.8% of the students), 106 (70.2%) identified as bisexual.

Table 2 shows the results of the questionnaires considering the gender identity and sexual orientation of the participants. The percentage of medical students with depression symptoms was 63.4%, with 39.5% of the students experiencing severe symptoms. Women exhibited a greater percentage of depressive symptoms than men, along with a greater percentage of moderate to severe symptoms. The percentage of depression symptoms was greater among bisexual students than among gay and lesbian students.

In our sample, all medical students had moderate to severe anxiety symptoms. There were no significant differences observed between gender and sexual orientation concerning trait and state anxiety.

**Table 1. Participant characteristics.** Total number of students (400).

| Characteristics of participants | N (%) |
|---|---|
| Gender identity | |
| Men | 249 (62.2%); |
| Women | 151 (37.8%); |
| Sexual Orientation | |
| Lesbian or gay | 257 (64.3%) |
| Bisexual | 143 (35.8%) |
| Course period | |
| 1°/2° | 139 (34.8%) |
| 3°/4° | 160 (40.0%) |
| 5°/6° | 101 (25.3%) |
| School administration | |
| Public | 248 (62.0%) |
| Private | 152 (38.0%) |
| School location | |
| Capital | 197 (49.3%) |
| Small City | 203 (50.7%) |

SD = standard deviation

Women presented higher overall quality of life scores than men did ($P$ = .043). Compared with men and gay or lesbian students, women and bisexual students reported poorer self-perceptions of academic performance ($P$ = .001 and $P$ = .004, respectively). With respect to gender and sexual orientation, no significant differences were observed in the resilience scores. In terms of internalized stigma, bisexual students demonstrated greater stigma scores than lesbian or gay students did.

Table 3 shows the relationships between the severity of depression symptoms and other variables. The percentage of depressive symptoms was highest among students who lived alone (61.1%), and 32.5% of this group reported moderate to severe symptoms. This was followed by students who lived with family and those who lived with friends.

Depression symptoms were lower in students involved in sports associations (53.4%) than in those involved in other associations (70%) and those not involved in any association (63.6%, $P$ = .026).

Compared with nonusers, students who utilized psychiatric medications presented a greater percentage of depression symptoms. Furthermore, students who consumed alcohol occasionally displayed fewer symptoms of depression than those who used alcohol frequently or never.

No significant differences were observed in depressive symptoms when age, religious status, tuition scholarship status, or graduation period were evaluated.

Table 4 shows the relationships between symptoms of depression and mental health scores. Students with a poorer self-perception of academic performance presented a greater percentage of depression symptoms. Severe symptoms were observed in 75% of the students in the "bad or very bad" group, 25.8% of the students in the "regular" group, and 17.5% of the students in the "good or great" group ($P$ < .001).

Severe trait anxiety was present in 72.7%, 56.9%, and 26.4% of the individuals with severe, moderate, and mild depression symptoms, respectively ($P$ < .001). For individuals with severe, moderate, or mild depression symptoms, the mean quality of life scores were 6.1, 7.1, and 8.0, respectively ($P$ < .001). The mean scores for quality of life during medical school were 4.7, 5.9,

**Table 2. Results of the questionnaires by gender and sexual orientation.**

| | Total | Gender Identity | | | Sexual orientation | | |
|---|---|---|---|---|---|---|---|
| | | Men | Women | P | Lesbian or gay | Bisexual | P |
| General quality of life (mean ± SD) | 7.2 (1.5) | 7.1 (1.0) | 7.4 (1.3) | .043 | 7.1 (1.3) | 7.2 (1.4) | .600 |
| QoL during medical school (mean ± SD) | 5.9 (1.7) | 6.0 (1.4 | 5.7 (1.3) | .170 | 5.9 (1.6) | 5.9 (1.4) | .800 |
| Perception of academic performance | | | | | | | |
| Very Bad + Bad | 28 | 10 (4.0%) | 18 (11.9%) | .001 | 12 (4.7%) | 16 (11.2%) | .004 |
| Regular | 155 | 89 (35.7%) | 66 (43.7%) | | 92 (35.8%) | 63 (44.1%) | |
| Good + Great | 217 | 150 (60.2%) | 67 (44.4%) | | 153 (59.5%) | 64 (44.8%) | |
| Beck's Depression inventory | | | | | | | |
| Total Score (Mean ± SD) | 13.1 (8.3) | 12.5 (7.8) | 14.4 (8.8) | .015 | 12.5 (8.3) | 14.3 (8.1) | .032 |
| No symptoms | 148 (37%) | 103 (41.4%) | 45 (29.8%) | .045 | 106 (41.2%) | 42 (29.4%) | .051 |
| Mild | 153 (38.3%) | 92 (36.9%) | 61 (40.4%) | | 94 (36.6%) | 59 (41.3%) | |
| Moderate/Severe | 99 (24.7%) | 54 (21.7%) | 45 (29.8%) | | 57 (41.3%) | 42 (29.4%) | |
| Trait anxiety | | | | | | | |
| Total score (Mean ± SD) | 49.7 (4.5) | 49.5 (4.3) | 50.0 (4.8) | .281 | 49.7 (4.5) | 49.7 (4.6) | .965 |
| Mild | 0 | 0 | 0 | .303 | 0 | 0 | .677 |
| Moderate | 91 (22.8%) | 131 (52.6%) | 71 (47.0%) | | 132 (51.4%) | 70 (49.0%) | |
| Severe | 309 (77.3%) | 118 (47.4%) | 81 (53%) | | 125 (49.6%) | 73 (51.0%) | |
| State anxiety | | | | | | | |
| Total score (Mean ± SD)) | 51.9 (3.3) | 51.8 (3.2) | 51.9 (3.5) | .660 | 52.0 (3.3) | 51.6 (3.3) | .224 |
| Mild (N, %) | 0 | 0 | 0 | .326 | 0 | 0 | .902 |
| Moderate (N, %) | 202 (50.5%) | 61 (24.5%) | 30 (19.9%) | | 58 (22.6%) | 33 (23.1%) | |
| Severe (N, %) | 198 (49.5%) | 188 (75.5%) | 121 (80.1%) | | 121 (80.1%) | 110 (76.9%) | |
| Resilience (Mean ± SD) | 70.42 (14.02) | 71.04 (3.37) | 69.4 (5.03) | .260 | 70.79 (3.47) | 69.76 (4.99) | .482 |
| Internalized Stigma (Mean ± SD) | 20.25 (8.45) | 20.36 (8.57) | 20.09 (8.27) | .767 | 19.27 (7.72) | 22.00 (9.39) | .004 |

SD = standard deviation.

and 6.7, respectively ($P < .001$). Additionally, the mean scores for resilience were 59.5, 69.3, and 78.9, respectively ($P < .001$), and the mean scores for internalized stigma were 22.5, 20.2, and 18.8, respectively ($P = .004$).

Table 5 presents the results of the multinomial logistic regression analysis between depression symptoms and various factors, including gender, sexual orientation, medical course-related quality of life, trait anxiety, resilience, and internalized stigma. The data revealed that being a man was associated with a lower likelihood of depression than being a woman, with no correlation after adjusting for sexual orientation. Additionally, being a lesbian or gay person was linked to a reduced likelihood of depression symptoms compared with being bisexual, with no correlation after adjusting for gender. Other factors associated with a lower likelihood of depression include living with friends instead of living alone or with family, participating in sports associations, having better general and medical course-related quality of life, having less trait anxiety, being more resilient and having less internalized stigma.

## Discussion

In support of our hypothesis, we observed a notably high percentage of depression symptoms among gay, lesbian and bisexual students. This percentage of depression symptoms surpasses that identified in the general population of Brazilian medical students (41.3%) [45] and the global population of medical students (27.3%) [1]. Furthermore, we noted a greater proportion of students with severe symptoms (39.5% versus 31.0% among general medical students) [45].

**Table 3. Participants according to the results of the beck inventory for depression.**

| | No symptoms | Mild | Moderate/Severe | P |
|---|---|---|---|---|
| Total | 148 (37%) | 153 (38.3%) | 99 (24.8%) | |
| Age (years) (Mean ± SD) | 22.6 (3.0) | 22.6 (2.4) | 22.8 (2.6) | .822 |
| Residential status | | | | |
| With Family | 38 (33.9%) | 42 (37.5%) | 32 (28.6%) | .016 |
| Alone | 33 (28.9%) | 44 (38.6%) | 37 (32.5%) | |
| With Friends | 77 (44.3%) | 67 (38.5%) | 30 (17.2%) | |
| Financial Assistance | 27 (29.0%) | 41 (44.1%) | 25 (26.9%) | .184 |
| Religious status | | | | |
| Practitioner | 62 (41.9%) | 53 (35.8%) | 33 (22.3%) | .293 |
| Nonpractitioner | 86 (58.1%) | 100 (39.7%) | 66 (26.2%) | |
| Association | | | | |
| Does not participate | 60 (36.4%) | 63 (38.2%) | 42 (25.5%) | .026 |
| Participates in sports association | 48 (46.6%) | 38 (36.9%) | 17 (16.5%) | |
| Participates in other associations | 40 (30.0%) | 52 (39.4%) | 40 (30.3%) | |
| Use of psychiatric medication | 29 (22.0%) | 52 (39.4%) | 51 (38.6%) | < .001 |
| Does not use psychiatric medications | 119 (44.4%) | 101 (37.7%) | 48 (17.9%) | |
| Drinks alcohol | | | | |
| Frequently | 24 (26.7%) | 42 (46.7%) | 24 (26.7%) | .018 |
| Sometimes | 111 (41.7%) | 90 (33.8%) | 65 (24.4%) | |
| No use | 10 (27.8%) | 16 (44,4%) | 10 (27,8%) | |
| Course period | | | | |
| Basic years (1st–2nd) | 52 (37.4%) | 48 (34.5%) | 39 (28.1%) | .557 |
| Clinic years (3rd–4th) | 60 (37.5%) | 67 (41.9%) | 33 (20.6%) | |
| Internship (5th–6th) | 36 (35.6%) | 38 (37.6%) | 27 (26.7%) | |

SD = standard deviation

Several factors may contribute to these findings, including the collegiate context of social belonging, longitudinal interactions with the same peer group, expectations of entering the job market, and internal competition, which may increase minority stress. A heavy workload in medical courses has also been considered a contributing factor, leading to increased reliance on peers and increased interactions with potential aggressors [46]. It is plausible that general stressors affect all medical students, as well as minority stressors specific to the LGB group, as outlined in the minority stress model. Additionally, there is a prevalence of internal and professional hierarchies and a heightened incidence of harassment in relationships between doctors and students [13].

Consistent with trends observed in other studies on the general population, our findings revealed a connection between depression and lower resilience [11, 47], increased anxiety [45], poorer quality of life [48, 49], and a less favorable perception of academic performance [6].

In contrast to studies in the general population of medical students, variables such as age, course period, college administration model, affirmative programs, and tuition scholarships did not differ in their impact on depression symptoms [45].

Surprisingly, no significant differences were identified between students residing in capital cities and those residing in smaller cities, in contrast to general population studies in which students living in capital cities presented more symptoms of depression [45]. We postulate that in culturally conservative small cities [50], the LGBTQIA+ population might be more

**Table 4. Results of the questionnaires according to depressive symptoms (Beck inventory).**

|  | No symptoms | Mild symptoms | Moderate/severe symptoms | P |
|---|---|---|---|---|
| Perception of academic performance |  |  |  |  |
| Good + Great | 96 (64.9%) | 83 (54,3%) | 38 (38.4%) | < .001 |
| Regular | 50 (33.8%) | 65 (42.5%) | 40 (40.4%) |  |
| Bad + Very Bad | 2 (1.4%) | 5 (3.3%) | 21 (21.2%) |  |
| Quality of Life |  |  |  |  |
| General Quality of Life (Mean ± SD) | 8.0 (1.0) | 7.1 (1.3) | 6.1 (1.6) | < .001 |
| Quality of Life during Medical School (Mean ± SD) | 6.7 (1.3) | 5.9 (1.5) | 4.7 (1.7) | < .001 |
| Trait Anxiety |  |  |  |  |
| Mild | 0 | 0 | 0 | < .001 |
| Moderate | 109 (73.6%) | 66 (43.1%) | 27 (27.3%) |  |
| Severe | 39 (26.4%) | 87 (56.9%) | 72 (72.7%) |  |
| State Anxiety |  |  |  |  |
| Mild | 0 | 0 | 0 | .583 |
| Moderate | 34 (23.0%) | 31 (20.3%) | 26 (26.3%) |  |
| Severe | 114 (77.0%) | 122 (79.7%) | 73 (73.7%) |  |
| Resilience (Mean ± SD) | 78.9 (9.4) | 69.3 (11.9) | 59.5 (14.8) | < .001 |
| Internalized Stigma (Mean ± SD) | 18.8 (6.6) | 20.2 (7.4) | 22.5 (10.5) | .004 |

vulnerable to minority stress, countering the observed trend of reduced depression among medical students in these regions.

Additionally, our study revealed that gay, lesbian and bisexual students living alone or with their families experienced more depression symptoms than those living with friends, a distinction not observed in the general population of medical students [45, 51]. One possible explanation for this observed difference is that gay, lesbian and bisexual students living with their families have more family conflicts because they are part of the LGB community than when they choose to live with friends.

Another notable difference was observed in alcohol consumption; students who occasionally consume alcohol experience fewer depression symptoms than those who abstain or consume excessive alcohol, possibly reflecting a marker of socialization.

Participation in sports associations was associated with lower levels of depression than engaging in other associations, which usually include political activities. Although sports themselves are known to offer protection against depression [52], we suggest that belonging to a group may have a protective effect against minority stress, while political activities expose individuals to the issues inherent to it.

Our study revealed that gay, lesbian and bisexual populations had lower quality of life scores than Brazilian medical students in general (7.2 vs. 7.9 for general quality of life and 5.9 vs. 6.5 for quality of life during medical school)51. These lower scores may be linked to the lower resilience scores and higher prevalence of depression observed in this population [27].

To our knowledge, no other published study conducted with medical students has reported that 100% of medical students have moderate to severe anxiety symptoms [53]. Additionally, in the general population, women typically experience more anxiety than men. However, in our population, a ceiling effect was observed, diminishing distinctions between lesbian or gay students and bisexual students. In line with other studies, women in our study experienced greater levels of depression symptoms than men, although the sex difference was less pronounced than that reported in previous studies conducted in Brazil [45]. The impact of minority stress related to sexual orientation may tend to mitigate gender disparities, a trend also reflected in quality-of-life scores.

**Table 5. Odds ratios (with 95% confidence intervals) of the associations between symptoms of depression (Beck inventory) and other variables, crude and adjusted models.**

|  | **Mild Symptoms** | **Moderate/Severe Symptoms** |
|---|---|---|
| **Crude Models** |  |  |
| Gender Identity |  |  |
| Women | Reference | Reference |
| Men | 0.86 (0.41–1.06; P = .086) | 0.52 (0.31–0.89; P = .017) |
| Sexual Orientation |  |  |
| Bisexual | Reference | Reference |
| Lesbian or gay | 0.63 (0.39–1.02; P = .062) | 0.54 (0.32–0.92; P = .023) |
| Residential status |  |  |
| Living with family | 1.27 (0.74–2.20; P = .392) | 2.16 (1.15–4.06; P = .017) |
| Living alone | 1.53 (0.88–2.68; P = .134) | 2.88 (1.53–5.41; P = .001) |
| Living with friends | Reference | Reference |
| Association |  |  |
| Sports associations | 0.69 (0.42–1.14; P = .146) | 0.43 (0.23–0.81; P = .009) |
| Other associations | Reference | Reference |
|  | Mild Symptoms | Moderate/Severe Symptoms |
| Quality of Life |  |  |
| Quality of life in general | 0.47 (0.37–0.60; P = .000) | 0.30 (0.22–0.39; P = .000) |
| Quality of life during medical school | 0.65 (0.55–0.77; P = .000) | 0.41 (0.33–0.50; P = .000) |
| Trait Anxiety |  |  |
| Severe trait anxiety | Reference | Reference |
| Moderate trait anxiety | 0.27 (0.17–0.44; P = .000) | 0.13 (0.08–0.24; P = .000) |
| Resilience | 0.92 (0.90–0.94; P = .000) | 0.87 (0.84–0.90; P = .000) |
| Internalized Stigma | 1.02 (0.99–1.06; P = .87) | 1.06 (1.03–1.11; P = .001) |
| **Adjusted Models (a)** |  |  |
| **Gender Identity** |  |  |
| Bisexual | Reference | Reference |
| Lesbian or gay | 0.72 (0.41–1.29; p = 0.268) | 0.69 (0.31–1.31; p = 0.257) |
| Home |  |  |
| Living with family | 1.28 (0.74–2.22; p = 0.376) | 2.19 (1.16–4.14; p = 0.016) |
| Living alone | 1.50 (0.86–2.62; p = 0.159) | 2.77 (1.47–5.23; p = 0.002) |
| Living with friends | Reference | Reference |
| Association |  |  |
| Sports associations | 0.68 (0.41–1.13; p = 0.133) | 0.42 (0.23–0.79; p = 0.007) |
| Other associations | Reference | Reference |
| Quality of Life |  |  |
| Quality of life in general | 0.43 (0.34–0.56; p<0.000) | 0.27 (0.20–0.36; p<0.000) |
| Quality of life in medical course | 0.65 (0.55–0.77; p<0.000) | 0.41 (0.33–0.51; p<0.000) |
| Trait Anxiety |  |  |
| Severe trait anxiety | Reference | Reference |
| Moderate trait anxiety | 0.27 (0.17–0.44; p<0.000) | 0.14 (0.08–0.24; p<0.000) |
| Resilience | 0.92 (0.90–0.94; p<0.000) | 0.87 (0.84–0.90; p<0.000) |
| Internalized Stigma | 1.02 (0.98–1.06; p = 0.130) | 1.06 (1.02–1.09; p = 0.001) |
| **Adjusted Models (b)** |  |  |
| **Sexual Orientation** |  |  |
| Women | Reference | Reference |
| Men | 0.79 (0.45–1.39; p = 0.410) | 0.64 (0.34–1.21; p = 0.171) |

(*Continued*)

**Table 5.** (Continued)

| | Mild Symptoms | Moderate/Severe Symptoms |
|---|---|---|
| Home | | |
| Living with family | 1.27 (0.74–2.20; p = 0.389) | 2.17 (1.15–4.09; p = 0.017) |
| Living alone | 1.51 (0.86–2.64; p = 0.150) | 2.82 (1.49–5.32; p = 0.001) |
| Living with friends | Reference | Reference |
| Association | | |
| Sports associations | 0.66 (0.40–1.10; p = 0.109) | 0.40 (0.22–0.77; p = 0.005) |
| Other associations | Reference | Reference |
| Quality of Life | | |
| Quality of life in general | 0.46 (0.36–0.59; p<0.000) | 0.29 (0.22–0.38; p<0.000) |
| Quality of life in medical course | 0.64 (0.54–0.77; p<0.000) | 0.40 (0.33–0.50; p<0.000) |
| Trait Anxiety | | |
| Severe trait anxiety | Reference | Reference |
| Moderate trait anxiety | 0.27 (0.16–0.44; p<0.000) | 0.13 (0.07–0.24; p<0.000) |
| Resilience | 0.92 (0.90–0.94; p<0.000) | 0.87 (0.84–0.89; p<0.000) |
| Internalized Stigma | 1.02 (1.00–1.05; p = 0.218) | 1.05 (1.02–1.09; p = 0.002) |

The group of students with no symptoms was used as a reference. The numbers are arranged in the following order: odds ratio (confidence interval and p). a: adjusted by gender identity; b: adjusted by sexual orientation.

Our data concerning anxiety symptoms should be interpreted with caution. A review of 388 published studies of the State-Trait Anxiety Inventory (STAI-T), Knowles and Olatunji concluded that although this inventory was developed to measure an individual's tendency to experience anxiety symptoms, there is a strong relationship between trait anxiety measured by the STAI-T and depressive disorders. They proposed that the STAI-T be considered a nonspecific measure of negative affectivity rather than trait anxiety per se [54, 55].

Unlike studies of the general population, where a higher prevalence of depression in women has been linked to poorer perceptions of quality of life and academic performance [50, 56], in our study, we only observed poorer perceptions of academic performance. On the other hand, as in the general population, there was no discernible difference in resilience scores between genders [11].

According to the trajectory of publications on sexual minorities [18, 57, 58], bisexual individuals presented a greater percentage of depression symptoms than their gay/lesbian counterparts. This observation aligns with greater internalized stigma and a poorer perception of academic performance in the bisexual group.

After we adjusted the logistic regression for gender and sexual orientation, it became evident that there were no significant differences between men and women or between lesbian/gay students and bisexual students. This suggests that within the studied population, the distinctions observed between men and women and between lesbian/gay individuals and bisexual individuals specifically manifest in the intersectional relationship with bisexual women. This subgroup appears to be the most vulnerable to minority stress.

Our study has several strengths, including the number of participants, the use of a multicentric approach, and the use of validated instruments widely employed in previous studies. The study was unable to have a higher number of participants because of the snowballing design, as students started to invite colleagues who had already been invited to participate in the study. While the data were exclusively collected within the Brazilian territory, we may reasonably infer that similar trends could be extrapolated to other countries.

Since it is a study with a nonprobability sample, the sample's representativeness is a limitation. There could be characteristics of the students who decided to answer the survey. It is possible that people with more symptoms decided to participate in the study, which may have influenced the high percentages of depression and anxiety observed. However, the percentage of invited people who agreed to participate was high (55.6%).

Another limitation of our study is that the cutoff values of the depression inventory have not been established for sexual and gender minority populations. Interestingly, Borgogna et al. demonstrated that, in college students, gender and sexual minorities respond differently than cisgender/heterosexual men and women to anxiety and depression questionnaires and that the cutoff values for anxiety and depression may differ [59].

## Conclusions

Gay, lesbian and bisexual medical students exhibit higher depression symptoms, with bisexual individuals emerging as particularly vulnerable. Depression, anxiety, internalized stigma, a poorer perception of academic performance, lower quality of life during medical school, lower resilience, and reduced socialization lead to a detrimental cycle of distress. Our study underscores the need for targeted and interdisciplinary institutional policies to foster mental health and social inclusion. These initiatives should focus on enhancing coping mechanisms, resilience, and overall quality of life while actively working to curb discrimination and prejudice within medical schools, with a special emphasis on supporting LGBTQIA+ students.

## Supporting information

**S1 Data. Data used in the manuscript.**
(XLSX)

## Author Contributions

**Conceptualization:** Felipe Scalisa, Renata Kobayasi, Milton A. Martins, Patricia Tempski.

**Data curation:** Milton A. Martins.

**Formal analysis:** Felipe Scalisa, Renata Kobayasi, Patricia Tempski.

**Funding acquisition:** Milton A. Martins.

**Investigation:** Felipe Scalisa, Renata Kobayasi, Milton A. Martins.

**Methodology:** Milton A. Martins, Patricia Tempski.

**Writing – original draft:** Felipe Scalisa, Renata Kobayasi, Milton A. Martins, Patricia Tempski.

**Writing – review & editing:** Felipe Scalisa, Renata Kobayasi, Milton A. Martins, Patricia Tempski.

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
