## [Decision Letter · Decision Letter 0]

26 Aug 2024

PMEN-D-24-00289

Mental health and minority stress in LGBTQIA+ medical students

PLOS Mental Health

Dear Dr. Martins,

Thank you for submitting your manuscript to PLOS Mental Health. Sorry that the review process takes longer than usual as we had to invite a third reviewer to provide additional perspectives for the first round of review. After careful consideration of the reviewers' feedback, we feel that it has merit but does not fully meet PLOS Mental Health’s publication criteria as it currently stands. Therefore, we invite you to submit a revised version of the manuscript that addresses the points raised during the review process.

EDITOR: 

When revising the manuscript, I recommend using Reviewer 2's comments as a reference point, as they provided comprehensive suggestions for improving various sections of the manuscript.

Reviewer 3 has serious concerns about the measures used, and I suggest reviewing the citations provided and explaining how you have addressed these limitations within the Brazilian context.

I would also appreciate seeing a paragraph in the introduction that elucidates the Brazilian context to better ground the findings of this study.

We look forward to receiving your revised manuscript.

Kind regards,

Kyle Tan, PhD

Academic Editor

PLOS Mental Health

**Comments to the Author**

1. Does this manuscript meet PLOS Mental Health’s publication criteria? Is the manuscript technically sound, and do the data support the conclusions? The manuscript must describe methodologically and ethically rigorous research with conclusions that are appropriately drawn based on the data presented.

Reviewer #1: Yes

Reviewer #2: No

Reviewer #3: No

2. Has the statistical analysis been performed appropriately and rigorously?

Reviewer #1: Yes

Reviewer #2: No

Reviewer #3: No

3. Have the authors made all data underlying the findings in their manuscript fully available (please refer to the Data Availability Statement at the start of the manuscript PDF file)?

Reviewer #1: Yes

Reviewer #2: No

Reviewer #3: No

4. Is the manuscript presented in an intelligible fashion and written in standard English?

Reviewer #1: Yes

Reviewer #2: Yes

Reviewer #3: Yes

5. Review Comments to the Author

Reviewer #1: Dear Authors,

Congratulations on your manuscript. It addresses a significant topic in scientific literature and provides valuable insights to the field. The manuscript is well-written and effectively elucidated with clear and informative tables. Additionally, it meets the journal's criteria comprehensively.

Reviewer #2: General comments

This is an interesting study that focuses on a population particularly susceptible to mental health problems, in whom it is necessary to have more information. However, the article has important limitations that unfortunately diminish its quality.

In general, the authors analyzed many variables with depressive symptoms, while it may be an objective for an article, is not entirely consistent with the title and objective stated in the study. For example, the title includes the concept of minority stress and in the Results Section I see only one variable related to minority stress (internalized stigma) and this has little prominence in the study.

If the study did not consider trans students because there were only 3 participants and did not include asexual, queer or intersex students, the article should be limited to the sexual orientations actually considered in the study. Therefore, including LGBTQIA+ in the title would be incorrect because not all of those sexual orientations, gender, and sex were represented.

A minor correction is that I believe the authors use the concept of “sexuality” when they mean “sexual orientation.” For example, the title of Table 2.

Despite all the comments I point out below, I encourage the authors to improve their article to resubmit it for evaluation to an academic journal.

Abstract

It is recommended to improve the reporting of the results by making a synthesis and a better integration of the results. This will improve their comprehension.

It is recommended to include more information about the method in the abstract regarding the participants, instruments, and analyses used.

I believe it is not correct to state that the correlation of a prevalence will be examined since the authors conducted a study at the individual level. Rather, I believe that the authors seek to examine the association between mental health variables (e.g., depressive symptoms or symptom severity) with sociodemographic variables, quality of life, resilience, and internalized stigma. Prevalence is an epidemiological indicator with respect to a group/population, therefore, the variable examined is not prevalence, but symptoms/severity of depression.

Introduction

The first paragraph of the article corresponds to the relevance of the study, therefore, I recommend including it at the end of the introduction, before the objective.

An important aspect of this study is that it was conducted in Brazil. Given the importance of the social context in the minority stress model, I suggest incorporating briefly what is the mental health situation or other important information regarding the context in which LGBTQIA+ health university students live in Brazil, or of the Brazilian LGTBQIA+ population in general in case there is no data from the same population studied in the article.

Materials and Methods

Study design and sample

The authors should mention the type of study used, which in this case would be a cross-sectional study.

They should also include the number of participants, the recruitment method, number of participating universities, average age, inclusion/exclusion criteria, and any other data relevant to understanding who the participants are.

I do not think it is necessary to repeat the variables measured in this section as they are mentioned in more detail below.

Ethics statement

The ethics statement could be incorporated as part of the procedure, which the authors called Data Collection.

Data collection

I am not sure what the authors mean by “We considered the sample representative when the indications became repetitive and reached a plateau”. Since it is a study with a non-probability sample, the representativeness of the sample is a limitation of their study. There could be characteristics of people that influence who decide to answer the survey and this could constitute a bias. For example, it is possible that people with more symptoms decided to participate in your study and this may have influenced the high percentages of depression and anxiety found.

Instruments

I am not sure what is the reason why the authors assessed gender identity by two different variables (lines 154-156).

Also, I wonder what the usefulness of the body mass index variable in the context of the study is and how it was measured. Was it a self-report of BMI?

There are variables called sociodemographic that are not precisely sociodemographic information (alcohol habits, BMI, academic performance, etc.).

It is not clear if the internal consistencies of the instruments were obtained in the sample or are from other studies. It is recommended to incorporate the internal consistency of the instruments obtained in the sample.

Is the symptom severity classification of the BDI and IDATE based on cut-off points? It is important to include this information.

It is recommended to include the references of the Brazilian adaptations of the instruments reported by the authors. It is not clear if they are already validated instruments or if they adapted them for this study.

Were the items to assess quality of life created or adapted by the authors or are they based on measures used in other studies? It is recommended to include this information and include the items. I do not think that readers are familiar with the concept of “quality of life in medical school”, so I believe that more information could be positive.

I suggest that the instruments include what high scores on the scales mean, so that people can better understand the results. For example, higher scores on the RS-14 scale indicate greater resilience.

Analysis

I am not sure what they mean by “According to Beck's Depression Inventory, the chi-square test was used”.

I think it would be simpler to mention that t-test was used for continuous variables and chi2 for categorical variables.

More information is needed on how they performed the analyses. For example, Table 4 compares quality of life with 3 levels of depression. T-test and logistic regression use binary variables, so it is not clear to me how they obtained this p-value. Also, in Table 5 there are different models and some with adjustment variables. It is not clear how these models were constructed, were all the models adjusted? what value of the sexual orientation variable is “adjusted for sexuality” and of gender is “adjusted for gender”?

Results

What happened to the trans students, were they excluded from the analyses?

Table 1 contains in the variable gender identity results for bisexual sexual orientation. I recommend including the information on the number of bisexual students by gender only in the text as it is confusing the way they included it in the table. It is the only variable that has a stratification.

Given that it is a study with a small, non-random and, therefore, non-representative sample, I am not convinced by the use of the concept of prevalence. With the study conducted, could we be sure that 63.4% and 100% of the university population in LGB health in Brazil have depression and anxiety, respectively? I think it would be more appropriate to refer to these indicators as percentages.

In Table 2 for the continuous variables SD and 95CI are used, it is suggested to use only one and that it be consistent throughout the study.

Table 3 has repeated information with Table 2 (e.g., symptoms of depression by gender and sexual orientation).

Since this is a cross-sectional study, caution should be exercised with the interpretation of the results as a protective or risk factor.

Maybe it is a translation error, but the authors mention “correlation” in some cases which I assume they mean “relation”. This is because they do not mention in the analyses that they have performed correlations.

It is possible that the authors did not find statistically significant results given that the depression variable was recoded as a 3-category variable. This results in fewer people within each model, which clearly may influence lower statistical power. Other alternatives are to use the continuous depression variable or to recode it into a binary variable for the regression models.

Discussion

The first paragraph should be a general summary of the results, without repeating percentages. This helps to have a general idea of the results of the study.

Regarding the use of the concept of prevalence, it is clearer in the discussion that 63.4% could be overestimated compared to what is observed in the rest of the world. The same with having obtained 100% of moderate to severe anxiety symptoms.

It is not clear how the aspects mentioned in lines 315 to 317 contribute to minority stress. Moreover, no reference to other studies is made to affirm this. These aspects could also be factors that influence the mental health of non-LGB individuals, so I suggest improving this interpretation of the results.

In the articles, the tables are not usually referred to in the discussion.

The interpretation of the results between lines 386 and 393 is not clear. They mention that they found no differences, but then talk about that bisexual women were more vulnerable.

The authors mention the number of participants as a strength; however, this number could be considered low. Why do they consider it a strength?

The authors did not include limitations in the manuscript.

Implications are mentioned in the conclusion, but these usually go at the end of the discussion.

Reviewer #3: The authors aim to discern the role of various sociodemographic characteristics on the presentation of depression, anxiety, social support, and resilience within a sexual minority medical student sample. My main concerns pertain to the use of the State-Trait Anxiety Inventory (STAI) that has poor psychometric properties as shown by previous studies and is not a measure of anxiety, and the questionable validity of all other scales used in the current sample as demonstrated by a nonsignificant correlation between the Beck Depression Inventory and the STAI. While I appreciate the authors’ efforts to identify contributors of minority stress within sexual minority medical students, empirical models built upon questionable measures are unlikely to produce knowledge that will generalize to those outside of the current study sample. I realize it is a tall order to ask authors to validate each of these measures and deviates from the main purpose of the study but it is crucial that psychometric soundness of all measures used be properly ascertained and demonstrated.

Please see below for some of my suggestions that I offer in the hope that they may help increase the quality of this paper.

1. Gender versus sex assigned at birth: The authors use the terms referring to gender versus sex assigned at birth interchangeably, which should be revised so that the concept they are referring to is clear and consistent throughout the paper. In their sociodemographics table, they report percentages of people who identified as “male” versus “female” but then mention differences between men and women. “male” and “female” do not refer to the social construct of gender and the paper needs to be revised to reflect this.

Please see the following guide on choosing terminology pertaining to gender provided by APA: https://apastyle.apa.org/style-grammar-guidelines/bias-free-language/gender

The authors allude that they gave “other” gender and “other” sexual orientation in the Instruments section, but the tables imply they only provided “male” and “female” and “gay/lesbian” and “bisexual” as options. I think it would be helpful to revise and make it clear what categories were offered within the table or at least mention in a footnote, considering that people may respond differently based on options provided. For example, most researchers will treat being nonbinary, genderfluid, genderqueer, and gender non-conforming as discrete from being transgender, which is not the case and gender expansive individuals can identify as binary transgender, nonbinary and transgender, nonbinary and not transgender, etc. However, offering discrete categories would mask this intersection of identities within gender.

2. The authors should consider that many people who identify as plurisexual but not necessarily bisexual may have chosen bisexual because no other choices were given, which is often the case for individuals who identify as pansexual.

3. Did the authors exclude the transgender and pansexual identifying individuals they mention at the beginning of the Results section? Or were they just excluded from group comparisons?

4. The limits of using the State-Trait Anxiety Inventory (STAI): It is a well-known fact that the STAI, especially the trait anxiety subscale, has serious psychometric limitations. Studies show that this measure does not assess for anxiety, but rather negative affectivity. Please refer to the following articles that make this very clear (as well as other articles that also further evidence this):

Bados, A., Gómez-Benito, J., & Balaguer, G. (2010). The state-trait anxiety inventory, trait version: does it really measure anxiety?. Journal of personality assessment, 92(6), 560-567

Knowles, K. A., & Olatunji, B. O. (2020). Specificity of trait anxiety in anxiety and depression: Meta-analysis of the State-Trait Anxiety Inventory. Clinical psychology review, 82, 101928.

Accordingly, the authors will need to make considerable revisions in their manuscript so that the STAI is not referred to as an anxiety scale and thus, they could not have examined “anxiety” within this sample with it, unless they had other measures available (e.g., GAD-7).

5. Relatedly, authors state “No significant correlation was found between depression and state anxiety” which is surprising given previous findings that suggest the opposite would hold: “Furthermore, anxiety and depressive symptom severity were similarly strongly correlated with the STAI-T (mean r = .59 – .61).” (Knowles & Olatunji, 2020). This makes me wonder if there are psychometric issues with the scales in this study’s sample, which would not be surprising. The authors support their use of the measures they opted for based on Cronbach’s alpha (which itself is a highly limited measure of internal consistency – for more information, please see: Rodriguez, A., Reise, S. P., & Haviland, M. G. (2016). Evaluating bifactor models: Calculating and interpreting statistical indices. Psychological methods, 21(2), 137.), but that does not speak towards construct validity – are these measures actually measuring depression and anxiety in this sample? I realize ascertaining this requires a lot more work and may be challenging to determine, but their results already suggest that the models they are running here may be inherently flawed, which warrants some investigation. It should not be surprising if the measures do not work the way they do in cisgender heterosexual samples, and authors can find that and show that, that itself is quite interesting and a valuable contribution of scientific knowledge for the field of minority mental health research. There is already some evidence that this may be the case:

Borgogna, N. C., Brenner, R. E., & McDermott, R. C. (2021). Sexuality and gender invariance of the PHQ-9 and GAD-7: Implications for 16 identity groups. Journal of Affective Disorders, 278, 122-130.

I wouldn’t be surprised if these limitations of measurement apply to the remainder of all scales utilized in the current study, which makes it difficult to further commentate on the discussion since all interpretations are assuming that all constructs were validly and reliably measured.

6. Authors also do not provide how they determined severity of depression or “anxiety” using these scales – more information needs to be provided on how cutoffs were determined as well as support for those cutoff values.

7. Some edits are needed to make it clear the limits of generalizability; for example, the authors state “The prevalence of depression symptoms among LGBTQIA+ medical students was 63.4%, with 39.5% of the students experiencing severe symptoms”; rather, they should make it clear that this was within THEIR sample as this may make it appear that this is a sweeping generalization of all LGBTQ+ medical students.

6. PLOS authors have the option to publish the peer review history of their article (what does this mean?). If published, this will include your full peer review and any attached files.

**Do you want your identity to be public for this peer review?** For information about this choice, including consent withdrawal, please see our Privacy Policy.

Reviewer #1: No

Reviewer #2: No

Reviewer #3: No

---

## [Decision Letter · Decision Letter 1]

31 Oct 2024

PMEN-D-24-00289R1

Mental health in gay, lesbian and bisexual medical students

PLOS Mental Health

Dear Dr. Martins,

Thank you for re-submitting your manuscript to PLOS Mental Health. Reviewer 1 has provided some additional comments following the earlier revision that your team made. After careful consideration, we feel that it has merit but does not fully meet PLOS Mental Health’s publication criteria as it currently stands. Therefore, we invite you to submit a revised version of the manuscript that addresses the points raised during the review process.

We look forward to receiving your revised manuscript.

Kind regards,

Kyle Tan, PhD

Academic Editor

PLOS Mental Health

Journal Requirements:

Reviewers' comments:

Reviewer's Responses to Questions

**Comments to the Author**

1. If the authors have adequately addressed your comments raised in a previous round of review and you feel that this manuscript is now acceptable for publication, you may indicate that here to bypass the “Comments to the Author” section, enter your conflict of interest statement in the “Confidential to Editor” section, and submit your "Accept" recommendation.

Reviewer #2: All comments have been addressed

2. Does this manuscript meet PLOS Mental Health’s publication criteria? Is the manuscript technically sound, and do the data support the conclusions? The manuscript must describe methodologically and ethically rigorous research with conclusions that are appropriately drawn based on the data presented.

Reviewer #2: Partly

3. Has the statistical analysis been performed appropriately and rigorously?

Reviewer #2: Yes

4. Have the authors made all data underlying the findings in their manuscript fully available (please refer to the Data Availability Statement at the start of the manuscript PDF file)?

Reviewer #2: No

5. Is the manuscript presented in an intelligible fashion and written in standard English?

Reviewer #2: Yes

6. Review Comments to the Author

Reviewer #2: I appreciate the opportunity to review this article. In this version of the manuscript, the authors have considerably improved the manuscript and the reporting of results, appropriately integrating the comments made in the first revision.

Some minor improvements to the paper are mentioned below.

General comment

The aim of the study is different from what was done in the study, as the authors only analyze factors associated with depression among college students. As it is currently written it is implied that they will also analyze the relationship between some factors and anxiety.

Abstract

Information about the measures used and the analysis performed is needed.

Keywords

It may be useful to include LGB as a keyword.

Introduction

In the first paragraph it says “prevalence rate”, but I think the authors mean “prevalence”.

When the acronym LGBTQIA+ is mentioned for the first time, the meaning of each letter should be explained.

In the third paragraph the authors mention that LGBTQIA+ identity is a risk factor. I suggest modifying that sentence because it could be understood that the higher prevalence of mental health problems is due to an individual aspect of LGBTQIA+ people. For example, it could be mentioned that LGBTQIA+ people in different contexts exhibit worse mental health outcomes than their heterosexual and cisgender peers.

Results

Second paragraph, an SD has a % symbol.

Third paragraph, I recommend when mentioning table 2 to say something similar to the title of table 2. As written in the manuscript it is not clear.

It is mentioned that “Women exhibited higher overall quality of life scores than men did (P=.043).”. However, in the table it is mentioned that this difference is not statistically significant.

In some parts of the results section the information in the tables is repeated in the text (for example, the means and percentages). I recommend improving the reporting of results by avoiding repeating information when it is not necessary, since the information is already in the table.

I recommend avoiding the use of terms such as protective factor or risk factor, as this is a cross-sectional study. Concepts such as protective/risk factor may imply causal relationships.

It says “logistic regression”, but the authors used multinomial logistic regression. It repeats again in the discussion.

In the last paragraph, I did not understand what the authors mean by the sentence beginning with: “Other differences include living with friends instead of living alone or with family,…”.

Discussion

The first paragraph should be a summary of the main results of the article.

In the second paragraph, the authors mention different factors to explain the high presence of depression. It is not clear why these factors could increase minority stress. I personally believe that the results could account for the relationship between the presence of such general stressors (affecting all medical students) and minority stressors (specific to the LGB group) (as states the minority stress model). On the other hand, it may be that the general factors are particularly influential on the LGB group. For example, the stress associated with entering the job market might be higher in LGB individuals due to expectations of rejection. Therefore, I recommend making more explicit what the authors mean by these factors that affect medical students increase minority stress.

The authors mentioned: “Additionally, our study revealed that gay, lesbian and bisexual students living alone or with their families experienced more depression symptoms than did those living with friends, a distinction not observed in the general population of students45, 54.” This is an interesting result that the authors could discuss further. Could it be related to students living at home having more family conflicts due to being part of the LGB community? are students living with friends from high socioeconomic status, so they have less mental health problems?

The authors mentioned: “Participation in sports associations was associated with lower levels of depression than engaging only in political activities.” However, they did not identify students that participate in political activities (the variable is coded sports/other).

It is not clear what the authors mean by “in our study we only observed poorer perceptions of academic performance.”

It is not recommended to mention the tables in the discussion.

Tables

Table 1 needs a first row with headings.

P values could be included, even if they are not lower than 0.05. There is now one result that is greater than 0.05, but it is not marked as NS.

Table 2 in severe state anxiety for man, the numbers don’t match.

Table 4 has percentages by row (e.g. academic performance) and by column (trait anxiety), which is confusing. I recommend reporting percentages by row or column, but not the two.

Table 5. Multinomial logistic regression does not estimate OR, but RRR. The authors need to correct this. This table is very long. The authors could make adjustments to reduce it. For example, it is not necessary to include the “No symptoms” column. It is recommended to include the RRR, CI, and p value in different columns (not in the same cell). I think the authors changed the order of the adjusted models a and b, because it says adjusted for sexual orientation and this variable appears in the model. The same for the models adjusted for gender.

7. PLOS authors have the option to publish the peer review history of their article (what does this mean?). If published, this will include your full peer review and any attached files.

**Do you want your identity to be public for this peer review?** For information about this choice, including consent withdrawal, please see our Privacy Policy.

Reviewer #2: No

---

## [Editor Report · Decision Letter 2]

4 Dec 2024

PMEN-D-24-00289R2

Mental health in gay, lesbian and bisexual medical students

PLOS Mental Health

Dear Dr. Martins,

 Thank you for wholeheartedly taking onboard the reviewers' suggestions and making the changes accordingly. I had another thorough read of the manuscript, and I believe it’s close to the stage of final acceptance. There are just a few minor edits I would like the author to consider: 1.
Include subheadings in your abstract: Introduction, Methods, Results, and Conclusion.2.
Italicise subheadings in different parts of the paper to make it clear to the production editor that these are subheadings. 3.
Edit this sentence on page 9 to “The analyses were performed using IBM SPSS Statistics Version 22.”4.
The character ‘P’ for p-value on page 10 should be italicised.5.
Edit this sentence on page 13 to “It is plausible that there are general stressors affecting all medical students, as well as minority stressors specific to the LGB group, as outlined in the minority stress model.”6.
Edit this sentence on page 16 to “Another limitation of our study is that the cut off values of the depression inventory have not been established for sexual and gender minority populations.” There was a typo with ‘stablished’. 7. The odds ratios in Table 5 should be rounded to two decimal places.8.
I recommend another thorough read-through of the article by a professional editor, if possible. 

We look forward to receiving your revised manuscript.

Kind regards,

Kyle Tan, PhD

Academic Editor

PLOS Mental Health
---

## [Editor Report · Decision Letter 3]

30 Dec 2024

Mental health in gay, lesbian and bisexual medical students

PMEN-D-24-00289R3

Dear Dr. Martins,

Thank you for wholeheartedly taking on board the editors' and reviewers' suggestions. We hope you have found the revision process enjoyable. We are pleased to inform you that your manuscript 'Mental health in gay, lesbian and bisexual medical students' has been provisionally accepted for publication in PLOS Mental Health.

Best regards,

Kyle Tan, PhD

Academic Editor

PLOS Mental Health
